# Moderating Effect of Variables Associated with Positive Mental Health in a Mediation Model on Depression among College Students: Protocol for a Longitudinal Study

**DOI:** 10.3390/healthcare11121709

**Published:** 2023-06-11

**Authors:** Thanthai Thongprem, Rawich Deechaiya, Saranphong Chanachai, Angkul Ngaoratsamee, Nahathai Wongpakaran, Punjaree Wiriyacosol, Tinakon Wongpakaran

**Affiliations:** 1Faculty of Medicine, Chiangmai University, 110 Intawaroros Rd., T. Sriphum, A. Muang, Chiang Mai 50200, Thailand; thanthai_thong@cmu.ac.th (T.T.); rawich_dee@cmu.ac.th (R.D.);; 2Department of Psychiatry, Faculty of Medicine, Chiang Mai University, 110 Intawaroros Rd., T. Sriphum, A. Muang, Chiang Mai 50200, Thailand; nahathai.wongpakaran@cmu.ac.th (N.W.); punjaree.w@cmu.ac.th (P.W.)

**Keywords:** borderline personality disorder symptom, attachment, anxiety, depression, perceived stress, interpersonal problems, negative family climate, character strength, inner strength, four sublimes’ states of mind

## Abstract

Background: Depression is significant mental health issue among university students and can decrease their ability to succeed academically. While various factors associated with mental health problems have been identified, researchers are increasingly exploring the role of positive mental health, such as character strengths and inner strengths, in relation to mental health problems. Objective: This study aims to build upon previous research by examining the influence of positive mental health on the mediation model of depression among students enrolled at Chiang Mai University. Method/Design: This study will employ an observational and longitudinal study that collects data from undergraduate students at Chiang Mai University during the 2023–2024 academic year. The primary outcome of this study will be depression. In the mediation models, insecure attachment and negative family climate will serve as predictors, while borderline personality symptoms will be the mediator. Positive mental health, including character strengths, inner strengths, and resilience, will be tested for moderation effects on the mediation models. Data will be collected at three intervals, with a three-month gap between each collection period. Discussion and Conclusion: This study will provide insights into the positive and negative mental health outcomes of university students in Chiang Mai. Through comprehensive analysis, this study intends to offer valuable insights into both positive and negative mental health outcomes among university students in Chiang Mai. Moreover, by employing a longitudinal approach, it aims to establish a more robust understanding of the causal relationships between positive mental health, predictors, mediators, and depression. The study’s limitations will also be discussed.

## 1. Introduction

### 1.1. Depression among University Students

There has been a significant increase in the prevalence of mental health problems among university students [1], especially during COVID-19. Depression is a significant public health issue and a common problem worldwide, often leading to suicide [2]. College and university students face a higher risk of experiencing depression, due to various factors, including academic pressure, exams, inadequate rest, lack of exercise and proper nutrition, stress, anxiety, and chronic illness [3].

### 1.2. Psychosocial Factors Related to Depression

While the factors mentioned above are linked to the development of depression, psychosocial factors, particularly those related to the family, are widely recognized as powerful predictors within this age group. Among the challenges faced by these students, separation from their families stands out as a significant source of stress. This separation can lead to feelings of loneliness and a sense of disconnection as they adjust to the unfamiliar environment of their new academic setting. The absence of familiar support systems and the need to adapt to a different social and cultural context can further exacerbate feelings of isolation and contribute to difficulties in adjusting to their new surroundings. This emotional and psychological strain underscores the importance of providing appropriate support and resources to help these students cope with the challenges associated with separation and facilitate their adjustment to the new environment [4].

In addition to the current perception of family support, it is crucial to consider how individuals perceive their upbringing and the quality of their parental relationships. Good nurturing relationships from caregivers or parents are essential for developing secure attachment in infants. Much research has revealed that growing up in an adverse family environment may bring about personality pathology. When infants experience unresponsiveness and rejection from caregivers, as well as the loss of an attachment figure, their attachment patterns may become insecure [5]. 

During infancy, individuals who have insecure attachments may have a higher likelihood of developing impaired social, psychological, and neurobiological functioning, which can make them more vulnerable to developing psychopathology, compared to children who formed secure attachments during this critical period [6]. There are notable similarities between insecure attachment and borderline personality disorder (BPD) in both empirical research and theoretical analysis. Studies revealed that the disorganized or dismissing style was strongly related to borderline personality traits [7]. During emerging adulthood, BPD becomes prominent and is associated with adverse and persistent psychological symptoms, e.g., depression, self-harm, and recurrent suicidal ideation, that are greater among young people [8]. Insecure attachment derived from a negative family climate is also found to be related to borderline personality and depression [9]. The role of BPD symptoms in mental health problems has been illustrated in addition to depression, for example, non-suicidal self-injury [10] and suicidality [11]. Researchers have discovered that borderline personality disorder symptoms serve as a mediator of the relationship between stress and depression [12].

### 1.3. Role of Social Support on Depression

During the phase of emerging adulthood, the perception of social support plays a vital role in shaping an individual’s mental well-being. Research has consistently shown that the presence or absence of adequate social support from both family and friends significantly influences mental and behavioral outcomes. Insufficient social support has been associated with an increased risk of various mental health issues, including anxiety, depression, and even internet addiction.

When individuals perceive a lack of support from their immediate social network, they may experience heightened feelings of distress, loneliness, and a diminished sense of belonging. These negative emotions can contribute to the development or exacerbation of mental health conditions. The absence of a supportive network to turn to during times of stress or difficulty may hinder effective coping mechanisms and exacerbate psychological symptoms.

Therefore, recognizing the critical role of social support during this developmental stage is essential. Promoting and fostering strong social connections, both within the family and with friends, can serve as a protective factor against the development of mental and behavioral problems. Providing individuals with accessible and meaningful support systems can help mitigate the negative impacts of social isolation and promote better mental well-being during emerging adulthood [4,13,14,15]. Numerous studies have demonstrated the role of perceived social support as a mediator and a moderator for depression.

### 1.4. Role of Some Positive Attributes as Protecting Factors for Depression

To identify university students’ mental health conditions, more comprehensive psychosocial factors should be taken into account. A previous study presented a mediation model of depression in medical students, highlighting the impact of adverse family climate on depression. Borderline personality disorder symptoms were identified as a mediator in this relationship [16]. Additionally, the study authors found that the relationship was moderated by inner strengths, such as meditation and perseverance. The presence of symptoms of BPD played a significant mediating role in the connection between a negative family environment and depression. The indirect effect was significant, and the depression scores were explained up to 49% when moderators were added to the existing models. This study, however, was limited in terms of causal relationships, as it was a cross-sectional study. In addition to the frequency of meditation and level of perseverance, perceived social support has long been studied as a moderator for depression in much research [17,18]. Research also demonstrated that life satisfaction, resilience, emotional regulation, flourishing, and beliefs regarding adversity have significant moderating influences on the relationship between stress and depression. This suggests that these positive psychological attributes can mitigate the adverse effects of stress on depression [19]. Along the same lines, there are still many aspects of character strengths and inner strengths that remain unexplored or have yet to be thoroughly examined, such as gratitude, self-regulation, generosity, and loving-kindness.

### 1.5. Aims

The primary objective of this study is to investigate the moderating role of positive mental health at both individual and social levels on the relationship between various factors, including insecure attachment, adverse family climate, borderline personality disorder symptoms, and depression. Specifically, the study aims to examine the potential impact of the following positive psychological attributes on the mediation models of depression: (1) resilience, (2) gratitude, (3) character strength, (4) inner strength (e.g., generosity), (5) perceived social support, and (6) sublime or four immeasurable (e.g., loving-kindness). The researchers hypothesize that each of these positive attributes will exert a significant moderating effect on the relationships between the negative predictors (insecure attachment and adverse family climate) and negative mental health outcomes (borderline personality disorder symptoms and depression). They propose that these positive attributes will act as buffers or protective factors, influencing the strength and direction of these relationships.

By exploring the specific influence of positive mental health attributes on the proposed mediation models, the study seeks to deepen our understanding of the complex interplay between these variables. The researchers anticipate that positive psychological attributes will play a crucial role in shaping the relationship between negative predictors and negative mental health outcomes through their impact on borderline personality disorder symptoms. This investigation aims to shed light on the potential pathways through which positive mental health attributes may mitigate the adverse effects that insecure attachment, adverse family climate, and borderline personality disorder symptoms may have on depression. By uncovering these moderating effects, the study has the potential to contribute to the development of interventions and strategies that leverage positive mental health attributes to promote better mental well-being and resilience in individuals facing challenging circumstances (Figure 1).

## 2. Materials and Methods

### 2.1. Study Design

This study is an observational, longitudinal study of Chiangmai University students. Data will be collected at three different points in time for each participant, first at baseline (0 months), then at a three-month follow-up, and finally, at a 6-month follow-up.

Participants: Chiang Mai University students. Currently (13 November 2021), according to the database on Chiangmai University’s official website, there are 22 faculties at Chiangmai University.

### 2.2. Study Population

The participants will be undergraduate students from Chiang Mai University.

Inclusion criteria: students who are:(1)Studying for a bachelor’s degree at Chiangmai University in the 2022–2023 academic year;(2)20–30 years old.

Exclusion criteria: students who

(1)Have problems understanding the Thai language;(2)Have underlying mental disorders, such as bipolar disorder, who can be in danger after filling out the questionnaire;(3)Have drug abuse and suicidal thinking;(4)Did not completely fill out the questionnaire.

### 2.3. Procedure and Participant Invitation

This study will employ a longitudinal design, including an online survey of participants who are undergraduate students at Chiang Mai University. To recruit participants, the convenience and snowball sampling techniques will be employed through various channels, including flyers, public websites, and social media platforms, such as Facebook and LINE. A small compensation will be offered to each participant (THB 100 or USD 3.03). To gather the information, we will develop an online questionnaire using Microsoft Forms.

### 2.4. Measurements

The Screening Instrument for Borderline Personality Disorder (SI-Bord) is comprised of six questions that correspond to the DSM-5 diagnostic criteria for borderline personality disorder. These questions assess the presence of abandonment avoidance, interpersonal relationship instability, identity disturbance, suicidal and self-harm behaviors, and affective instability. It is a 5-point Likert scale, ranging from never like me (0) to exactly like me (4). The total score ranges from 0 to 24. The higher score represents more BPD symptoms or traits. Cronbach’s alpha was 0.75 [20].

The Experiences in Close Relationships-Revised (ECR-R) questionnaire is an updated version of the ECR questionnaire [21]. The ECR-R items were selected from the same item pool as the ECR questionnaire but use techniques based on the item response theory. Both instruments aim to measure individual differences in attachment-related anxiety (i.e., the degree of insecurity or security regarding the availability and responsiveness of romantic partners) and attachment-related avoidance (i.e., the degree of discomfort or security in relying on others for emotional closeness). The ECR-R comprises 10 items, with a 7-point rating scale. The Thai version of the ECR-R demonstrated a Cronbach’s alpha of 0.84 for the anxiety subscale and 0.85 for the avoidance subscale [22].

The Family Climate Questionnaire (FCQ) is a unidimensional outcome measure comprising 64 items and was designed to assess emotions related to family climate. The FCQ measures eight emotional dimensions, with four representing positive climates (i.e., warm/relaxing, assisting/helping, vivacious/fun/enjoyable, and sensitive/empathic) and four representing negative climates (i.e., overinvolved/controlling, ignoring/distant, chaotic/disorganized, and lonesome) [16]. Each dimension consists of eight questions, with a 5-point Likert scale (1 = not at all, 2 = some, 3 = often, 4 = almost always). The FCQ segregates questions for mothers and fathers. Negative family climate (NFC) is derived from the combination of negative dimension scores for parents, whereas positive family climate (PFC) is derived from the combination of positive dimension scores for parents. The FCQ was tested among a sample of 240 university students and was demonstrated to be unidimensional, with a Cronbach’s alpha of 0.97 for both PFC and NFC [23].

PhuSeG scale, a 10-item composite scale of prudence, humility, self-regulation, and gratitude, was derived from the four scales measuring prudence, humility, self-regulation, and gratitude. It was developed by Wongpakaran N and Wongpakaran T. The five-item Likert-scale is numbered from 1, very unlike me, to 5, very similar to me. The score ranges from 10 to 50; the higher the score, the higher the positive psychology. Cronbach’s alpha was 0.85. The PhuSeg scale was shown to be unidimensional, and all items had fit statistics between 0.89–1.47 as recommended, with good person and item reliabilities (0.80 and 0.91, respectively) [24].

Gratitude inventory (GI) is a 20-item, unidimensional outcome measure designed to assess the character related to gratitude and is based on Theravada Buddhist culture. The GI has four4 Likert-type scale choices (strongly disagree, disagree, agree, and strongly agree). The GI was tested in a sample of 87 university students and was shown to be unidimensional. Cronbach’s alpha was 0.75.

The Four Immeasurables (FI) scale is a 10-item, unidimensional outcome measure designed to assess the character related to the sublime or the four immeasurables, which are loving, compassion, gladness, and equanimity. The FI item has five Likert-scale choices (‘Not at all’ like me, ‘Very little’ like me, ‘Somewhat’ like me, ‘To a great extent’ like me). FI was tested in a sample of 87 university students and shown to be unidimensional. Cronbach’s alpha was 0.72.

The I-SBI-7 (Inner Strength-Based Inventory version 7) is a 50-item questionnaire that assesses 10 positive behavioral characteristics inspired by the 10 perfections from Buddhist teachings, including generosity, perseverance, wisdom, precept, meditation, equanimity, loving-kindness, determination, tolerance, and truthfulness. Each characteristic has five items, and responses are recorded on a 10-point scale. Mean scores are calculated for each item. The person reliability of the I-SBI-7 was 0.86, based on Rasch analysis [25].

The 21-item outcome inventory (OI-21) is a self-rating questionnaire that assesses four prevalent mental health issues, including depression, anxiety, somatic symptoms, and interpersonal difficulties. The OI-21 comprises 21 questions, with a 5-point Likert scale ranging from 1 (never) to 5 (always). The Cronbach’s alpha value for the OI-21 was calculated as 0.92, indicating high internal consistency. Moreover, the OI-21 was well studied for its concurrent validity with various other measures [26].

The Perceived Stress Scale (PSS) is a 10-item self-report questionnaire that measures the level of stress experienced by the respondent during the past four weeks, using a 5-point Likert scale ranging from 0 (never) to 4 (very often). The total PSS score ranges from 0 to 40, with higher scores indicating greater perceived stress [27]. The Thai version of the PSS has good reliability and validity and has been widely used in both clinical and non-clinical samples. In this study, the PSS demonstrated a Cronbach’s alpha of 0.83 [28].

The revised Multidimensional Scale of Perceived Social Support (rMSPSS) is a 12-item scale that measures the perceived social support from significant others (SO), friends (FR), and family (FA) [29]. The response options are on a seven-point Likert scale, ranging from very strongly disagree (0) to very strongly agree (6). The higher scores indicate greater perceived social support. The revised Thai version of the rMSPSS has shown good psychometric properties. In this study, only the family subscale was used, which demonstrated a Cronbach’s alpha of 0.88 [30].

Thymometer is a short and single statement asking about perceived stress, depression, coping, and perceived social support. It has 4 questions, with 10 Likert responses ranging from 0 (not at all) to 10 (extremely). This scale shows acceptable validity through the significant correlations between the respective item and the full measurement [31].

See Table 1 for the summary of the measurements.

### 2.5. Statistical Analysis Plan

#### 2.5.1. Sample Size Calculation

The minimal number of required participants for the mediation model was calculated by using a formula from “Monte Carlo Power Analysis for Indirect Effects” from the link https://schoemanna.shinyapps.io/mc_power_med/, accessed on 10 June 2023. The sample size estimation was calculated by determining Type I error (alpha) at 0.05 and Type II error (beta, 1-power) at 0.20. According to Figure 1, the correlation coefficients between variables from previous studies yielded a minimum number of 250. However, based on the nature of a longitudinal survey, dropout was likely to occur. Therefore, the number of participants was expected to be higher to compensate for it, and the total sample used was 300.

#### 2.5.2. Statistical Analysis

Descriptive analysis (e.g., mean, standard deviation, percentages, etc.) will be used to analyze the participants’ demographic data, e.g., age, gender, and prevalence, or mean scores of attachment styles (secure or insecure), frequency of meditation practice, adherence to Buddhist moral precepts, and scores of positive (e.g., perceived social support) and negative (e.g., perceived stress, depression) mental health variables.

In comparing continuous data of the two groups, the student’s *t*-test/Mann–Whitney U test will be used (e.g., total scores of ECR-r in different genders), while Chi-square/Fisher’s exact test will be used for categorical data (e.g., sex and level of income).

Pearson’s correlation coefficient will be analyzed to test the correlation between continuous data if they are normally distributed, e.g., gratitude score and OI-depression scores. Spearman’s rank-order correlations will be applied to test the correlation between variables with ordinal data (e.g., number of precepts and OI-interpersonal). The biserial point correlations will be calculated when testing the correlation between two variables if one of the variables is dichotomous, e.g., secure and insecure attachment and perceived stress.

Mediation/Moderation: All significantly correlated variables will be tested in the hypothesis model to find the mediation and/or moderation effects among the variables. Negative family climate and attachment will be the predictors, borderline personality disorder symptoms will be the mediators, and the primary outcomes will be negative mental health factors, i.e., depression, anxiety, somatization, interpersonal difficulties, and perceived stress. Positive attributes, e.g., gratitude and inner strength, will be moderators.

For longitudinal data analysis, a parallel process latent growth curve model (PP-LGCM) will be utilized to investigate the moderated mediation effects over time.

PP-LGCM is a valuable approach used to evaluate longitudinal mediation effects within a structural equation modeling (SEM) framework. It offers an effective and efficient method to assess and interpret the causal relationships of longitudinal mediation effects, particularly in the context of intervention studies.

The SPSS version 22 will be used for analyzing the data. Amos version 18 will be used for the testing and analysis of mediation models. Mplus 8.4 will be used for PP-LGCM. *p* values < 0.05 will be considered significant.

### 2.6. Ethics Approval, Consent to Participate, Autonomy, and Confidentiality

The researchers in this study received training on research ethics, e.g., human subjects protection training, before conducting the research. An independent ethics committee approved this research. The researchers will consider safety and confidentiality as top priorities among the participants. The participants should volunteer to participate in this research. The data will not be shared with anyone. Only the management team can access all data. The researchers will follow Thailand’s PDPA (Personal Data Protection Act).

## 3. Discussion

The study constituted a relatively comprehensive evaluation of both positive and negative psychosocial factors on mental health among university students. Therefore, identifying and evaluating the mental health of university students is essential to warrant their success academically and their well-being.

As shown by considerable evidence, another critical issue related to students’ mental health is their childhood traumatic experiences and insecure attachments that lead to unstable personality traits, i.e., borderline personality. Recent findings have shown the impact of borderline personality disorder on students’ mental health and well-being. In contrast, some inner strengths moderate the relationship between borderline personality symptoms and depression. However, studies on the positive attributes or strengths that mediate or moderate the relationship between BPD symptoms and depression are still lacking. Our study will provide an opportunity to explore such positive mental health attributes as gratitude, resilience, self-regulation, meditation practice, and five precepts and their impacts on depression. The results yielded from the study will provide knowledge to implement interventions to promote mental well-being, to prevent depression, or to prevent an even more severe symptom—suicidality.

With some degree of the inclusive framework of psychosocial variables, unexplored hypotheses can be tested. Examples are whether resilience could mediate or buffer the relationship between stress and depression among students with insecure attachments or whether gratitude would moderate the relationship between symptoms of borderline personality disorder and depression. Would the four immeasurables mediate the relationship between anxiety and depression among students growing up in adverse family climates? What type of inner strengths are the best predictors for other positive character strengths or depression? Moreover, multiple positive attributes can also be examined.

Another advantage of the study is that the longitudinal design would allow us to obtain more robust evidence of the results. Because the limitations of the correlational studies found in many bodies of literature include employing a cross-sectional design, which precludes a causal inference from being established, the present study will help this gap by employing a longitudinal study among this cohort.

One potential limitation of the proposed research is that it relies entirely on self-report questionnaires, which may be susceptible to social desirability bias. Second, drop-out may occur when multiple rounds of data collection are employed. Finally, culture and religion might influence some results, especially those related to moral virtues.

### Theoretical Applications

The research findings can have clinical implications for Thai university students grappling with depression. A practical approach would involve initiating a screening process to identify those at risk for depression, based on factors such as adverse family climate, insecure attachment, and symptoms of borderline personality disorder. Upon identifying students who are at risk, it is recommended to implement interventions that promote positive mental health. These interventions may include practices such as meditation, loving-kindness exercises, perseverance training, honesty cultivation, equanimity practices, or gratitude-promoting activities. The study’s new knowledge will provide insights into the specific mediation models of depression related to different types of positive mental health. These positive mental health aspects will warrant further research in the future.

## 4. Conclusions

This longitudinal study aims to gather comprehensive information about positive and negative mental health outcomes among university students in Chiang Mai. By analyzing data collected over an extended period, the study will provide valuable insights into mental well-being trends and dynamics in this population. These insights will deepen our understanding of factors influencing mental health outcomes in university students and help identify early signs of negative mental health, such as anxiety, depression, and stress. The study will inform about the development of targeted interventions and evidence-based strategies to promote better mental health, addressing specific areas where students may struggle. Examining changes in mental health over time, the study will enable timely interventions and preventive measures to maintain and enhance mental well-being throughout the university experience. Ultimately, this longitudinal study will contribute to the development of comprehensive mental health support systems in universities, positively impacting the well-being and academic success of students in Chiang Mai.

## Figures and Tables

**Figure 1 healthcare-11-01709-f001:**
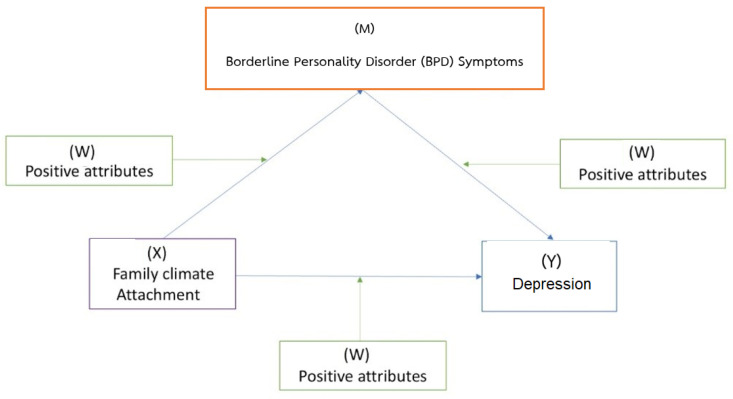
Conceptual model of the study. X = exogenous variable (predictor), Y = endogenous variable (outcome), W = moderator, and M = mediator.

**Table 1 healthcare-11-01709-t001:** Characteristics of the instruments.

Instrument	Aim in Accessing	Response Format	Number in Items	Recall Period	Internal Consistency
Screening Instrument for Borderline Personality Disorder (SI-Bord)	Level of borderline personality disorder symptoms	5	6	Currently	0.806
The Experiences in Close Relationships-Revised (ECR-R)	assess individual differences with respect to attachment-related anxiety	7	10	Currently	Attachment anxiety = 0.828 Attachment avoidance = 0.819
Family climate questionnaire (FCQ)	assess emotions related to family climate	4	64	Currently	PFC (positive family climate) = 0.901 NFC (negative family climate) = 0.869
PhuSeG scale	Level of character strength	5	10	Currently	0.795
Gratitude inventory (GI)	assess the character related to gratitude based on Theravada Buddhist culture	5	20	Currently	0.772
Four Immeasurables (FI) Scale	assess the character related to sublime or four Immeasurable	5	10	Currently	0.792
Inner Strength-Based Inventory (i-SBI)	Ten constructive conduct traits influenced by the ten perfections.	11	50	Previous week	0.912
The 21-item outcome inventory (OI-21)	measures four common mental health problems (depression, anxiety, somatic symptoms, and interpersonal difficulties	5	21	Previous week	0.921
PSS	Level of stress during the last four weeks	5	10	Past four weeks	0.734
Revised Multidimensional Scale of Perceived Social Support (rMSPSS)	Level of support by significant others, friends, and family	7	12	Currently	0.854
Thymometer	Level of perceived stress, depression, coping, and perceived social support	11	4	Past two weeks	N/A

## Data Availability

The datasets used and/or analyzed during the current study are available from the corresponding author on reasonable request.

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
