# Peer review of "Moderating Effect of Variables Associated with Positive Mental Health in a Mediation Model on Depression among College Students: Protocol for a Longitudinal Study"

_healthcare, 2023, doi:10.3390/healthcare11121709_

Round 1
Reviewer 1 Report
I am concerned there is not a results section. Although I may be overlooking something in the methods, the article cannot be considered without better understanding the statistical results.
I have highlighted some areas an made notes in the pdf in the first part of the article but am unwilling to continue until there is a section that discusses the results with more clarity.

There are several areas that need attention. I highlighted several in the introduction. Please define a mediator and moderator more specifically, and consider calling the meditation something else so one does not wonder if this is just a typo. What kind of meditation?
Author Response
Dear Editor,
We sincerely appreciate and are grateful to the reviewers for their valuable comments. We have thoroughly revised our manuscript based on their feedback. Below, you will find our detailed point-by-point response to each of the reviewers' comments.
Reviewer 1
Response: All of the highlighted texts in the PDF file, as indicated by the reviewers, have been revised (shown in blue or red color), except for Line 225, which states, "Descriptive analysis (e.g., mean, standard deviation, percentages, etc.) will be used to analyze the participants." We have maintained the future tense "will be" in this instance since the study has not yet been conducted.
Also, we have extended the content of the manuscript to meet the suggested minimum word count of 4000 words.
Reviewer 2
This is a study proposal that aims to investigate contributing factors and protective factors of depression among Thai university students. The inclusion of individual and social factors, as well as its longitudinal design are all merits of the study. Overall, I believe the study design could contribute to the literature. That said, I have several suggestions:
(1) The title of the study is too complex and missed the key feature of the study. “Moderation effect of the positive mental health on the mediation models of depression”? Broad terms make the imports of the title vague.
Response
We apologize for any confusion, but we would like to retain the same title for our study. Since we have multiple independent variables and mediators (positive mental health variables) to test, it will result in numerous mediation models. Therefore, it is not feasible to include a specific variable for independent variables and mediators in the title. However, we appreciate any suggestions you may have for a more specific title.
(2) The Introduction is weak. The authors started their introduction by mentioning “academic load”, but it did not show up in the model. So it seems to be intended as a background to the mental health problems of university students. In that case, academic load was not the only source of stress for university students; the introduction should not focus too much on this single aspect. Besides, the effects of attachment style, positive attributes, etc. on depression were well-documented, and it is then incumbent upon the authors to show the necessity of another study that seems to only replicate past research finding. One way to do this may be to highlight the population (i.e., Thai university students); or the authors may highlight some theoretical implication of their proposed model, as compared to past research.
Response: Thank you for your suggestion. We have added it as shown below.
Theoretical implications
Theoretical implications arise from research findings and can have significant implications for the development, refinement, or modification of existing theoretical perspectives and models concerning positive mental health and depression, particularly among Thai university students. They offer valuable insights that can help researchers and scholars expand their knowledge, generate new hypotheses, identify gaps in current theories related to positive mental health and depression, and guide future research endeavors in this field.
(3) This study was designed to be longitudinal, but the proposed statistical analysis plan was mostly for cross-sectional data. What is the plan for the longitudinal data analysis? How will the longitudinal data be used to analyze the moderated mediation model?
Response:
For longitudinal data analysis, a parallel process latent growth curve model (PP-LGCM) will be utilized to investigate the moderated mediation effects over time.
PP-LGCM is a valuable approach used to evaluate longitudinal mediation effects within a Structural Equation Modeling (SEM) framework. It offers an effective and efficient method to assess and interpret the causal relationships of longitudinal mediation effects, particularly in the context of intervention studies.
(4) In Figure 1, the criterion variable (Y) is stress. Should it be “Depression”?
Response. That’s correct. Our apologies. We have revised the figure.
(5) Minor grammar errors and stylistic errors exist, e.g., “Factors attributed to their mental challenges” (Line 41), “attributed” should be “contributed
Response: Thank you for your suggestion. We have corrected it.
Hopefully, we have satisfied the editor and the reviewers’ concerns. We are looking forward to hearing from you soon.
Best regards,
TW

Reviewer 2 Report
This is a study proposal that aims to investigate contributing factors and protective factors of depression among Thai university students. The inclusion of individual and social factors, as well as its longitudinal design are all merits of the study. Overall, I believe the study design could contribute to the literature. That said, I have several suggestions:
(1) The title of the study is too complex and missed the key feature of the study. “Moderation effect of the positive mental health on the mediation models of depression”? Broad terms make the imports of the title vague.
(2) The Introduction is weak. The authors started their introduction by mentioning “academic load”, but it did not show up in the model. So it seems to be intended as a background to the mental health problems of university students. In that case, academic load was not the only source of stress for university students; the introduction should not focus too much on this single aspect. Besides, the effects of attachment style, positive attributes, etc. on depression were well-documented, and it is then incumbent upon the authors to show the necessity of another study that seems to only replicate past research finding. One way to do this may be to highlight the population (i.e., Thai university students); or the authors may highlight some theoretical implication of their proposed model, as compared to past research.
(3) This study was designed to be longitudinal, but the proposed statistical analysis plan was mostly for cross-sectional data. What is the plan for the longitudinal data analysis? How will the longitudinal data be used to analyze the moderated mediation model?
(4) In Figure 1, the criterion variable (Y) is stress. Should it be “Depression”?
(5) Minor grammar errors and stylistic errors exist, e.g., “Factors attributed to their mental challenges” (Line 41), “attributed” should be “contributed”.
Overall, the English is fine, but minor issues exist.
Author Response

(The authors gave the same response as above.)

Round 2
Reviewer 1 Report
The edits have sufficiently addressed my concerns.
Author Response
Thank you very .much
Reviewer 2 Report
In the revised manuscript, the authors have addressed most of the problem. However, the added Theoretical Implication are general directions, not specific avenues of research that could benefit from the potential findings of the present study. As a result, the overall theoretical contribution still feels weak.
Author Response
Thank you for your concern. We have revised it accordingly.